# Secular Trends in Ablation Therapy for Graves’ Disease: An Analysis of a 15-Year Experience at a Tertiary Hospital in South Korea

**DOI:** 10.3390/jcm10081629

**Published:** 2021-04-12

**Authors:** Min Joo Kim, Ye An Kim, Sun Wook Cho, Su-jin Kim, Kyu Eun Lee, Young Joo Park, Do Joon Park, Bo Youn Cho

**Affiliations:** 1Department of Internal Medicine, Seoul National University Hospital, Seoul 03080, Korea; chorong24@gmail.com (M.J.K.); yeanin@gmail.com (Y.A.K.); yjparkmd@snu.ac.kr (Y.J.P.); djpark@snu.ac.kr (D.J.P.); bycho@cau.ac.kr (B.Y.C.); 2Department of Internal Medicine, Seoul National University College of Medicine, Seoul 03080, Korea; 3Seoul National University Hospital Healthcare System Gangnam Center, Seoul 06236, Korea; 4Department of Internal Medicine, Veterans Health Service Medical Center, Seoul 05368, Korea; 5Department of Surgery, Seoul National University Hospital, Seoul 03080, Korea; su.jin.kim.md@gmail.com (S.-j.K.); kyu.eun.lee.md@gmail.com (K.E.L.); 6Thyroid Center, Chung-Ang University Hospital, Seoul 06973, Korea

**Keywords:** ablation therapy, Graves’ disease, hyperthyroidism, radioactive iodine (RAI), surgery

## Abstract

Ablation therapy, such as radioactive iodine (RAI) therapy or thyroidectomy, is generally used as the second-line treatment for Graves’ disease (GD) in Asia. This study investigated changes in the clinical characteristics and outcomes of ablation therapies for GD over 15 years. Patients who underwent ablation therapy between 2001 and 2015 at a single tertiary hospital were included. Among the 10,991 GD patients treated over this 15-year period, 1357 (12.3%) underwent ablation therapy, and the most common reason was intractable GD. The proportion of patients who underwent any type of ablation therapy significantly decreased from 9.0% (2001–2005) to 7.7% (2011–2015). However, the proportion of patients who underwent surgery significantly increased from 1.1% (2001–2005) to 2.4% (2011–2015), and the proportion of patients who received ablation therapy due to suspected thyroid cancer increased from 5% to 13% over time. With a median follow-up duration of 6.2 years, remission was achieved in 86% and 98% of patients in the RAI and surgery groups, respectively, and these rates remained stable over time. In conclusion, although the proportion of patients who underwent ablation therapy for GD decreased during 15 years, the proportion of those who underwent surgery increased in association with the increased rate of suspected thyroid cancers.

## 1. Introduction

The therapeutic options for Graves’ disease (GD) include antithyroid drug (ATD) therapy, radioactive iodine (RAI) therapy using ^131^I, and thyroidectomy. Although several guidelines recommend the optimal indications for each treatment [1,2], the first-line treatment of choice for GD shows regional differences according to patients’ or doctors’ preferences. ATD therapy is the first-line treatment for European and Asian patients (84–97%), whereas RAI therapy is frequently chosen for American patients (59–75%) [3,4,5,6]. As in other Asian countries, ATD is the most frequently selected treatment for GD in South Korea. Surveys conducted by the Korean Thyroid Association (KTA) demonstrated that 81% of respondents in 1991 and 97% in 2012 chose ATD as the first-line treatment for GD [7,8]. Data from the Korean Health Insurance Review and Assessment Service (HIRA; 2006–2012) consistently showed that 91–98% of GD patients received ATD as the initial treatment [9]. Meanwhile, the initial remission rates for ATD therapy have been reported to be 40–50% even after 18 to 24 months of treatment [10,11], suggesting that more than half of GD patients need second-line treatment. However, ablation therapies such as RAI ablation or thyroidectomy were conducted in fewer than 10% of GD patients in South Korea, which is a considerably lower proportion than that of patients who require second-line therapy [9]. Indeed, a previous study showed that a subset of patients received ATD therapy three or more times with repeated recurrence of GD [11].

This study aimed to investigate the clinical characteristics and the long-term outcomes of GD patients who underwent ablation therapy by evaluating 15 years of clinical experience at a single referral hospital.

## 2. Materials and Methods

### 2.1. Subjects

For this retrospective study, 10,991 adult patients (20 years old or older) who had been treated for GD at a single tertiary referral hospital between 2001 and 2015 were identified. GD was defined based on laboratory data including the level of the thyroid stimulating hormone (TSH) receptor antibody and/or increased diffuse thyroid uptake of ^99m^Tc on radionuclide scintigraphy. Patients who underwent ablation therapy for GD were screened (*n* = 1504), and those who had received initial ablation therapy before 2001 (*n* = 128) were excluded. Thus, 1357 patients were included in the final analysis. Data on age, sex, the time of diagnosis, the reason for ablation therapy, the presence of Graves’ ophthalmopathy (GO), the pathologic results of surgery, and complications of surgery were obtained retrospectively through a review of patients’ electronic medical records. The presence of GO was identified based on the International Classification of Diseases, 10th revision code H06.2, and/or details of a visit to an ophthalmologist in our hospital for GO. The severity of GO was divided into mild and moderate-to-severe GO based on the European group on Graves’ Orbitopathy (EUGOGO) guideline [12]. The study was approved by the Institutional Review Board of Seoul National University Hospital (No. 1410-097-619).

### 2.2. Definition of the Clinical Outcomes of GD

Remission of GD was defined as the achievement of euthyroid or hypothyroid status following the withdrawal of ATD for more than 6 months. Immunologic remission was assessed by the achievement of the negative conversion of TSH receptor antibody after ablation therapy. TSH receptor antibody was measured using a radio-receptor assay kit (RSR Limited, Cardiff, UK) [11]. Permanent postoperative hypoparathyroidism was defined as persistent hypocalcemia with low parathyroid hormone levels (<10 pg/mL) for more than 1 year after surgery.

### 2.3. Statistical Analysis

Data are presented as the mean ± standard deviation. The chi-square test was used to evaluate the changes in ablation therapy over time. The characteristics of patients who received RAI therapy and surgery were compared, and categorical and continuous variables were analyzed using the chi-square test and the Student *t*-test, respectively. To compare the cancer characteristics of incidentally found thyroid cancer after surgery and thyroid cancer suspected before surgery, categorical and continuous variables were analyzed using the chi-square test and the Mann–Whitney *U* test, respectively. *p* values < 0.05 were considered to indicate statistical significance. All statistical analyses were performed using SPSS version 23.0 for Windows (IBM Corp., Armonk, NY, USA).

## 3. Results

### 3.1. Trends in Ablation Therapy for GD during the 15-Year Period

During the 15-year study period, a total of 10,991 patients were treated for GD, and 1357 (12.3%) patients underwent ablation therapy, including RAI therapy or thyroidectomy. All patients received ATD therapy as the first-line treatment of GD, except five patients who initially received RAI therapy because of liver function abnormalities. Figure 1 shows the changes in the proportion of patients who underwent each type of ablation therapy for 15 years, divided into 5-year periods. The proportion of patients who underwent ablation therapy significantly decreased from 9.0% (2001–2005) to 7.7% (2011–2015) (*p* for trend = 0.03; Figure 1a). The proportion of patients who underwent surgery significantly increased from 1.1% to 2.4%, whereas the proportion of patients who received RAI therapy significantly decreased from 8.0% to 5.3% over time (all *p* for trend < 0.01; Figure 1a). During the 15-year period, the extent of surgery changed. The proportion of subtotal thyroidectomy decreased, as it was replaced by total thyroidectomy (*p* for trend < 0.01; Figure 1b).

### 3.2. Changes in the Reasons for Ablation Therapy for GD during the 15-Year Period

Overall, the most common reason for receiving ablation therapy was intractable GD with or without large goiter (77%), followed by adverse events of ATD therapy (13%) and suspected thyroid cancer (10%). During the 15-year period, the proportion of patients who received ablation therapy for suspected thyroid cancer increased significantly from 5.3% to 13.0% (Figure 2a), and all of those patients underwent surgery. The proportion of the patients who underwent surgery due to suspected thyroid cancer significantly increased in the 2006–2010 period and subsequently decreased in the 2011–2015 period (Figure 2b). Meanwhile, the reasons for RAI therapy did not change over time (Figure 2c).

During the 15-year period, 10,986 GD patients initially received ATD treatment, and 154 patients (1.4%) then required ablation therapy due to severe adverse events. The proportion of these patients decreased from 1.1% (2001–2005) and 1.2% (2006–2010) to 0.7% (2011–2015) (*p* for trend = 0.047). The most common adverse event was toxic hepatitis (47%), followed by agranulocytosis (31%) and urticaria and/or skin rash (23%) (Appendix A). The incidence of toxic hepatitis and urticaria and/or skin rash was significantly higher in propylthiouracil (PTU) users than in methimazole users. RAI therapy was preferred over surgery in these patients (96% vs. 4%, *p* < 0.01).

### 3.3. Changes in the Clinical Characteristics of Patients during the 15-Year Period

Among the 1357 patients who underwent ablation therapy for GD, patients with a follow-up duration of less than 1 year (*n* = 156) were excluded, and the clinical characteristics of the remaining patients were analyzed. The mean age of the study subjects was 42 ± 13 years, and 71% were women. The time from diagnosis to ablation therapy was 5.1 ± 5.2 years. Over 15 years, the age at ablation therapy and the time from diagnosis to ablation therapy significantly increased (Table 1). The proportion of women of childbearing age (aged 20–44) significantly decreased to 47% (2001–2005), 40% (2006–2010), and 36% (2011–2015) (*p* for trend = 0.012). Although the prevalence of GO increased over time, the prevalence of moderate-to-severe GO did not differ (Table 1).

### 3.4. Clinical Characteristics of Patients According to the Treatment Modality

Next, the clinical characteristics of patients were analyzed according to treatment modality (RAI vs. surgery), and the surgery group was further divided according to the reason for surgery (the surgery for GD therapy, designated “surgery for GD”, and surgery for suspected cancer, designated “surgery for cancer”).

First, the RAI and surgery for GD groups were compared, since the reason for ablation therapy was similar in these two groups. The surgery for GD group showed a higher ratio of female sex (86% vs. 69%, *p* < 0.01), while the age at ablation therapy was similar between the two groups (Table 2). The proportion of women of childbearing age was significantly lower in the surgery for GD group than that in the RAI group (46% vs. 59%, *p* = 0.01). The time from diagnosis to ablation therapy in the surgery for GD group was significantly longer than that in the RAI group (6.9 ± 6.9 vs. 4.9 ± 4.8 years, *p* < 0.01). The prevalence of moderate-to-severe GO in the surgery for GD group was significantly higher than that in the RAI group (3.9% vs. 1.2%, *p* = 0.04). Among 12 patients with moderate-to-severe GO in the RAI group, seven patients with active GO received steroid treatment to prevent the worsening GO. A higher proportion of patients underwent ablation therapy due to adverse events of ATD in the RAI group than in the surgery for GD group (15% vs. 6%, *p* < 0.01; Table 2).

Next, the patients who underwent surgery for cancer were analyzed. Patients’ age at ablation therapy was older than that of the other two groups, and the female ratio was higher than that of the RAI group but similar to that of the surgery for GD group (Table 2). The time from diagnosis to ablation therapy and the follow-up duration were similar to those of the RAI group (Table 2).

### 3.5. Clinical Outcomes of Ablation Therapy

After a median follow-up duration of 6.2 years (range, 1.0–17.7 years), 86% (835/968) and 98% (229/233) of patients in the RAI and surgery groups, respectively, achieved remission. The remission rates for RAI therapy and surgery were similar during the three consecutive 5-year periods (Appendix A). As expected, surgery achieved a significantly higher remission rate (*p* < 0.01) despite a shorter follow-up duration (Table 2). After ablation therapy, 995 (83%) patients measured TSH receptor antibody at least once. Among 869 patients who achieved final remission, 390 (45%) patients have been proved to obtain the immunological remission, achieving negative conversion of TSH receptor antibody measurement.

In the surgery group, four patients experienced recurrence. All of them received subtotal thyroidectomy, and their mean time from surgery to recurrence was 6.3 years (range, 3.1–8.9 years). Meanwhile, there was no recurrence in patients who underwent total thyroidectomy. Serious surgical complications, including postoperative permanent hypoparathyroidism and vocal cord palsy, developed in 10 (4.3%) and two (0.9%) patients, respectively, and the rate of complications did not significantly differ according to whether patients underwent subtotal thyroidectomy or total thyroidectomy (4% vs. 8%, *p* = 0.26).

Postoperative pathological outcomes were also analyzed. Among the 127 patients in the surgery for cancer group, 110 (87%) had pathologically confirmed thyroid cancer, and 96% of 110 patients had differentiated thyroid cancer. Interestingly, 10% (11/106) of the patients in the surgery for GD group had incidentally found thyroid cancer after surgery. These incidentally found thyroid cancers were smaller (0.7 ± 0.7 vs. 1.3 ± 1.2 cm, *p* < 0.01) and had fewer extrathyroidal extensions (9% vs. 50%, *p* = 0.01) than those of the thyroid cancers suspected before surgery (Table 3).

## 4. Discussion

Among 10,991 GD patients, 1357 (12.3%) received ablation therapy, including RAI therapy or surgery, as the second-line treatment of GD. The most common reason for ablation therapy was intractable disease with or without a large goiter, followed by adverse events of ATD and suspicion of thyroid cancer. The proportion of GD patients who underwent RAI therapy significantly decreased during the 15-year period, whereas the proportion of patients who underwent surgery significantly increased in association with the increased rate of suspected thyroid cancer. After a median 6.2-year follow-up duration, the remission rate was significantly higher in the surgery group than in the RAI group, whereas both remission rates were markedly higher (86–98%) than in patients who received ATD therapy (52%), in accordance with the findings of a previous study [11].

During the 15-year period, the proportion of patients who underwent surgery within the ablation therapy group and the proportion of patients who underwent surgery for suspected thyroid cancer significantly increased. Compared to 2001–2005, the diagnosis of thyroid cancer increased in 2006–2015 with the technological evolution of high-resolution ultrasonography test in Korea [13,14]. Therefore, the proportion of RAI therapy has decreased relatively. An emerging question is whether cancer screening should be performed before selecting the treatment modality for GD patients. At present, clinical practice guidelines recommend no further cancer screening for all GD patients; however, cancer screening is performed with considerable frequency in real-world clinical practice. This study showed that 10% of the patients in the surgery for GD group had incidentally diagnosed thyroid cancers. Previous studies have reported that the pooled prevalence of incidental thyroid cancer was 7% (0.4–32.4%) [15]. Although 73% of these tumors were smaller than 0.5 cm, similar to the findings from autopsy data [16,17], 27% and 9% of them had multiplicity and lymph node metastasis, respectively. Thus, careful examinations may be needed when preparing for surgery. However, routine cancer screening for all GD patients is unfeasible due to the heterogeneous parenchymal changes in ultrasonographic imaging of these patients. In addition, the prevalence of thyroid cancer in GD patients has been reported to be similar to that in patients with euthyroid goiter or hyperthyroid patients without GD [15,18]. Moreover, the incidentally found small thyroid cancers in GD patients have shown a better prognosis than those found in the general population [19]. Therefore, further evidence is needed to conclusively answer this question.

This study demonstrated that ablation therapy was required in at least 12% of patients with GD after initial ATD treatment. In real-world practice, the selection of an optimal ablation therapy—that is, the choice between RAI therapy and surgery—is difficult except for patients with suspected thyroid cancer (surgery for cancer group). Although a long-term follow-up study of quality of life revealed no significant differences between RAI therapy and surgery [20], treatment should be carefully chosen in consideration of several factors, such as age, sex, goiter size, the presence of GO, comorbidities, and the individual patient’s socio-economic status. In this study, the surgery for GD group showed a higher proportion of female patients and a longer time from diagnosis to ablation therapy than the RAI group. However, the other characteristics did not differ significantly between the two groups. The presence of GO is an important factor in decision making. Because RAI therapy can induce and worsen GO [21], surgery is recommended over RAI therapy. Previous studies have reported that GO was associated with the decision to choose surgery over RAI therapy [22]. In this study, the prevalence of moderate-to-severe GO was also significantly higher in the surgery group than in the RAI group (3.9% vs. 1.2%), although the total prevalence of GO including mild form did not differ between groups (9.4% vs. 6.9%). The possible reason for the similar prevalence of GO may be that RAI therapy can be safely performed with steroid use in mild cases of GO [1,12]. Too low prevalence of GO may be another reason for the limited difference between groups. GO is less frequent and less severe in Asians than in Caucasians [23,24]. In addition, the presence of GO, especially mild cases, may have been undervalued and missed because it identified based on the diagnosis code and/or ophthalmology visit record in this study. In this study, patients with GO increased from 4% (2001–2005) to 10% (2011–2015), but the prevalence of moderate-to-severe GO did not differ over 15 years. In the 2010s, the rapid referral system between the thyroid center and ophthalmology team has been established in our hospital; thus, doctors became more interested in GO and referred to the ophthalmologist. Therefore, the increase in the prevalence of recorded GO reflects the increase in diagnosis of mild GO owing to the increase in doctor’s interest of it.

Of the patients who underwent ablation therapy, 13% underwent these treatments because of adverse events of ATD, similar to the proportion in previous reports [3]. Methimazole is the preferred ATD because of its effectiveness and favorable adverse event profile [6], and the use of methimazole has been increasing compared to that of PTU [3,9]; methimazole accounted for 55% in Korea and 60% in the United States, respectively [3,9]. Because methimazole is used more often, adverse events of methimazole are more frequently reported (Appendix A). Among the adverse events, toxic hepatitis was the most common cause for ablation therapy.

The remission rate of thyroidectomy (98%) was remarkably higher than that of RAI therapy (86%). A systematic review of 62 studies reported that thyroidectomy is more likely to be successful than RAI therapy [25]. In terms of surgical extent, the recurrence rate for total thyroidectomy is significantly lower than that for subtotal thyroidectomy, with an odds ratio of 0.1 [26,27]. In Korea, Sung et al. reported that the recurrence rate of GD was lower in patients who underwent total thyroidectomy than in those who underwent subtotal thyroidectomy (0% vs. 4.1%) [28], which is similar to the results of this study. However, the higher surgical complication rate of total thyroidectomy than that of subtotal thyroidectomy is an obstacle in choosing this procedure [26]. However, a meta-analysis of randomized clinical trials showed that the rate of permanent hypoparathyroidism and recurrent laryngeal nerve palsy after total thyroidectomy was not significantly different from that after subtotal thyroidectomy [27]. Furthermore, this study showed that total thyroidectomy had a significantly lower recurrence rate without increasing the rate of surgical complications. Therefore, total thyroidectomy is the recommended procedure of choice [1], and the preference for the surgical procedure has shifted from subtotal to total thyroidectomy over time (Figure 1b) [29].

Ablation therapy is indispensable for GD treatment because initial ATD treatment cannot achieve remission in all patients. Ablation therapy was performed in 12% of patients over 15 years. Although it is difficult to choose between RAI therapy and surgery, both types of ablation therapy showed satisfactory remission rates. Therefore, ablation therapy is actively recommended in patients who do not achieve remission after ATD therapy.

## Figures and Tables

**Figure 1 jcm-10-01629-f001:**
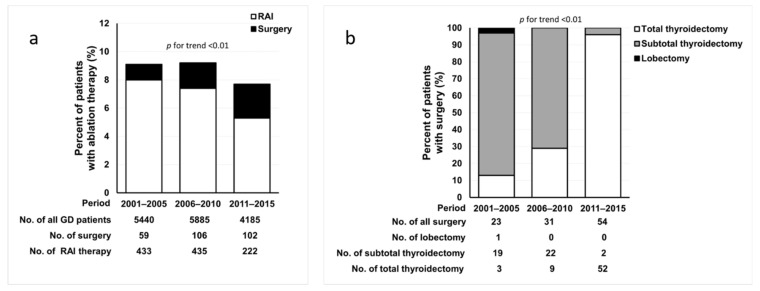
The trend of ablation therapy for Graves’ disease (GD) over 15 years. (**a**) Changes in radioactive iodine (RAI) therapy and surgery. (**b**) Changes in the extent of surgery.

**Figure 2 jcm-10-01629-f002:**
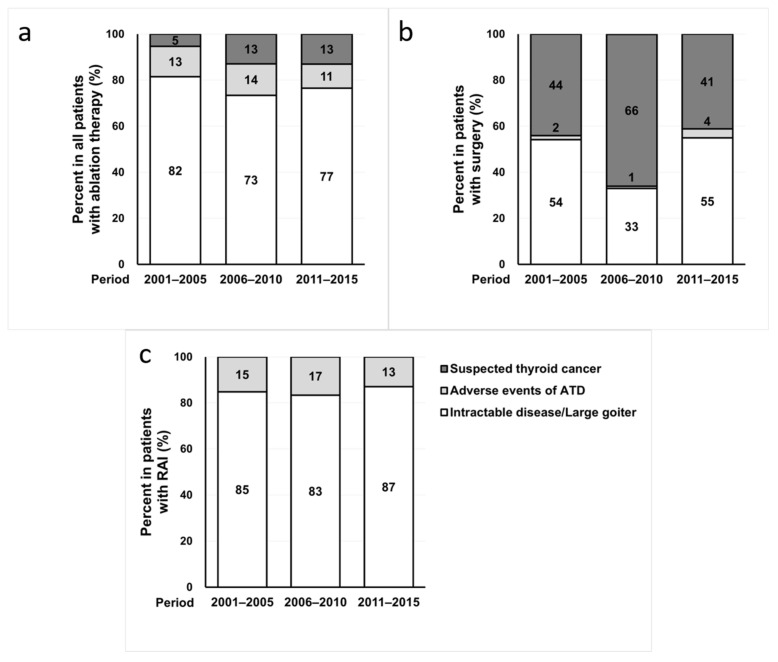
Changes in the reason for ablation therapy for Graves’ disease over time. (**a**) All patients, *n* = 10,991. (**b**) Patients who received surgery, *n* = 267. (**c**) Patients who received RAI therapy, *n* = 1090.

**Table 1 jcm-10-01629-t001:** Comparison of clinical characteristics for patients who underwent ablation therapy according to the period.

Characteristics	Total (*n* = 1201)	2001–2005 (*n* = 432)	2006–2010 (*n* = 474)	2011–2015 (*n* = 295)	*p* for trend
Age at ablation therapy (years)	42 ± 13	39 ± 12	42 ± 13	46 ± 14	<0.001
Sex, female, *n* (%)	855 (71)	309 (72)	322 (68)	224 (76)	0.300
Time from diagnosis to ablation therapy (years)	5.1 ± 5.2	4.4 ± 4.8	4.6 ± 4.8	6.6 ± 5.9	<0.001
GO, *n* (%)	89 (7)	17 (4)	43 (9)	29 (10)	0.002
Moderate-to-severe GO, *n* (%)	16 (1.3)	5 (1.2)	7 (1.5)	4 (1.4)	0.915

Data were presented as mean ± standard deviation or *n* (%). GO, Graves’ ophthalmopathy.

**Table 2 jcm-10-01629-t002:** Comparison of clinical characteristics among RAI and surgery groups.

Characteristics	RAI (*n* = 968)	Surgery for GD (*n* = 106)	Surgery for Cancer (*n* = 127)
Age at ablation therapy (years)	41 ± 13	39 ± 14	48 ± 14 ^a,b^
Sex, female, *n* (%)	663 (69)	91 (86) ^a^	101 (80) ^a^
Time from diagnosis to ablation therapy (years)	4.9 ± 4.8	6.9 ± 6.9 ^a^	4.4 ± 6.0
Follow-up duration (years)	7.2 ± 4.4	5.5 ± 3.3 ^a^	7.2 ± 3.1 ^b^
GO, *n* (%)	67 (7)	10 (9)	12 (9)
Moderate-to-severe GO, *n* (%)	12 (1.2)	4 (3.8) ^a^	0 (0)
Reason for ablation therapy, *n* (%)			
Intractable disease ± large goiter	820 (85)	100 (94) ^a^	
Adverse events of ATD	148 (15)	6 (6) ^a^	
Suspected thyroid cancer			127 (100) ^a,b^

Data were presented as mean ± standard deviation or *n* (%). RAI, radioactive iodine; GO, Graves’ ophthalmopathy; ATD, anti-thyroid drug. ^a^
*p* < 0.05 compared to the RAI group; ^b^
*p* < 0.05 compared to the surgery for GD.

**Table 3 jcm-10-01629-t003:** Comparison of cancer characteristics between incidentally found thyroid cancer after surgery and thyroid cancer suspected before surgery.

Characteristics	Incidental Cancer (Surgery for GD) (*n* = 11)	Suspected Cancer (Surgery for Cancer) (*n* = 110)	*p*
Mean age at surgery (years)	42 ± 13	48 ± 13	0.19
<55 years	9 (82)	74 (67)	0.50
≥55 years	2 (18)	36 (33)	
Sex, female, *n* (%)	10 (91)	86 (78)	0.46
Tumor type, *n* (%)			
PTC	9 (82)	97 (88)	0.46
FTC	2 (18)	9 (8)	
PDTC	0 (0)	2 (2)	
MTC	0 (0)	2 (2)	
Tumor size (cm), *n* (%)	0.7 ± 0.7	1.3 ± 1.2	<0.01
≤0.5 cm	8 (73)	25 (23)	<0.01
0.6–1.0 cm	1 (9)	39 (35)	
1.1–2.0 cm	1 (9)	29 (26)	
>2.0 cm	1(9)	17 (16)	
Multiplicity, *n* (%)	3 (27)	40 (36)	0.75
ETE, *n* (%)	1 (9)	54 (50)	0.01
LN metastasis, *n* (%)	1 (9)	25 (23)	0.45

Data were presented as mean ± standard deviation or *n* (%). PTC, papillary thyroid carcinoma; FTC, follicular thyroid carcinoma; PDTC, poorly differentiated thyroid carcinoma; MTC, medullary thyroid carcinoma; ETE, extra-thyroidal extension; LN, lymph node.

## Data Availability

Data sharing not applicable.

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
