# Peer review of "Secular Trends in Ablation Therapy for Graves’ Disease: An Analysis of a 15-Year Experience at a Tertiary Hospital in South Korea"

_jcm, 2021, doi:10.3390/jcm10081629_

Round 1

Reviewer 1 Report

Overall, this is a well-written and interesting article about the trends in ablation therapy for Graves’ disease over a 15-year period (2001-2015) at a single tertiary hospital. The analysis was sufficiently high powered, with 1,357 patients undergoing ablation therapy. The causes for ablation therapy were analyzed as well as the clinical characteristics in each group. This was compared across consecutive 5-year periods to give a sense of changes in practice over time.  

Few notes:

Abstract:

Line 24—clarify – both types of ablation therapy?

Results:

Line 142—“among next” ?

Line 167—“among after”?

Discussion:

Line 248: please provide reference for the assertion that GO is not as severe in asians vs. Caucasians

Please commend on why GO is seen more in 2011-2015 group for ablation compared to 2001-2005 (as shown in Table 1 data)

Figures/Tables:

Figure 1: please label Y-axis better, such as: “Percent of GD undergoing ablative therapy” or make the legend more descriptive

Table 1: add to the title: “for patients undergoing ablative therapy”

Table 2: title doesn’t make sense—according to the period? Seems like it is according to treatment modality instead?

Not sure Fig 3 is necessary. 

Table 3: median or mean age at surgery

Author Response

Reviewer #1

Abstract:

Comment 1> Line 24—clarify – both types of ablation therapy?

Response 1> The proportion of any type of ablation therapy (RAI therapy or thyroidectomy) decreased. We clarified it.

Revised manuscript, Abstract, Line 24:

The proportion of patients who underwent any type of ablation therapy significantly decreased from 9.0% (2001-2005) to 7.7% (2011-2015).

Results:

Comment 2> Line 142—“among next” ?

Response 2> It’s a mistake. We corrected it to “Next”.

Comment 3> Line 167—“among after”?

Response 3> It’s a mistake. We corrected it to “After”.

Discussion:

Comment 4> Line 248: please provide reference for the assertion that GO is not as severe in asians vs. Caucasians

Response 4> We added references [23, 24] for the sentence. Reference [23]: Tellez et al. reported that the risk for developing Graves’ ophthalmopathy in Europeans was 6.4 times higher than that in Asian (Clin Endocrinol (Oxf) 1992;36:291). Reference [24]: Khong et al reported that Caucasian ethnicity increased the risk for developing Graves’ ophthalmopathy compared to non-Caucasian (J Clin Endocrinol Metab 2016;101:2711).  

Revised manuscript, Discussion, Line 261:

Too low prevalence of GO may be another reason for the limited difference between groups. GO is less frequent and less severe in Asians than in Caucasians [23, 24].

Comment 5> Please commend on why GO is seen more in 2011-2015 group for ablation compared to 2001-2005 (as shown in Table 1 data)

Response 5> We appreciate the valuable comment. In the 2010's, the rapid referral system between thyroid center and ophthalmology team has been established in our hospital, thus doctors became more interested in GO, registered the diagnosis code, and referred to the ophthalmologist. We further analyzed the severity of GO through a review of patients’ medical records, and classified GO into two groups, mild and moderate-to-severe GO. We found that the prevalence of moderate-to-severe GO did not differ over 15 years. Therefore, the increase in the prevalence of recorded GO reflects the increase of diagnosis of mild GO owing to the increase of doctor’s interest of it.

Revised manuscript, Materials and Methods 2.1., Line 69:

The presence of GO was identified based on the International Classification of Diseases, 10th revision code H06.2 and/or details of a visit to an ophthalmologist in our hospital for GO. The severity of GO was divided into mild and moderate-to-severe GO based on the European group on Graves’ Orbitopathy (EUGOGO) guideline [12].

Revised manuscript, Results 3.3., Line 143:

Although the prevalence of GO increased over time, the prevalence of moderate-to-severe GO did not differ (Table 1).

Table 1. Comparison of clinical characteristics for patients who underwent ablation therapy according to the period.

Total

(n = 1,201)

2001-2005

(n = 432)

2006-2010

(n = 474)

2011-2015

(n = 295)

P

for trend

Age at ablation therapy (years)

42 ± 13

39 ± 12

42 ± 13

46 ± 14

<0.001

Sex, female, n (%)

855 (71)

309 (72)

322 (68)

224 (76)

0.300

Time from diagnosis

to ablation therapy (years)

5.1 ± 5.2

4.4 ± 4.8

4.6 ± 4.8

6.6 ± 5.9

<0.001

GO, n (%)

89 (7)

17 (4)

43 (9)

29 (10)

0.002

Moderate-to-severe GO, n (%)

16 (1.3)

 5 (1.2)

7 (1.5)

4 (1.4)

0.915

Data was presented as mean ± standard deviation or n (%).

GO, Graves’ ophthalmopathy.

Revised manuscript, Discussion, Line 265:  

In this study, patients with GO increased from 4% (2001-2005) to 10% (2011-2015), but the prevalence of moderate-to-severe GO did not differ over 15 years. In the 2010's, the rapid referral system between thyroid center and ophthalmology team has been established in our hospital, thus doctors became more interested in GO and referred to the ophthalmologist. Therefore, the increase in the prevalence of recorded GO reflects the increase of diagnosis of mild GO owing to the increase of doctor’s interest of it.

Figures/Tables:

Comment 6> Figure 1: please label Y-axis better, such as: “Percent of GD undergoing ablative therapy” or make the legend more descriptive

Response 6> We change the label of Y-axis to “Percent of patients with ablation therapy”.

Comment 7> Table 1: add to the title: “for patients undergoing ablative therapy”

Response 7> We changed the title of Table 1 to “Comparison of clinical characteristics for patients who underwent ablation therapy according to the period”

Comment 8> Table 2: title doesn’t make sense—according to the period? Seems like it is according to treatment modality instead?

Response 8> Thank you for the kind comment. We changed the title of Table 2 to “Comparison of clinical characteristics among RAI and surgery groups”

Comment 9> Not sure Fig 3 is necessary.

Response 9> We changed Figure 3 to Supplementary Figure 1.  

Comment 10>Table 3: median or mean age at surgery

Response 10> Mean age at surgery were presented in Table 3. We clarified it.

Revised Table3: Mean age at surgery

Reviewer 2 Report

The manuscript  investigated changes  in the clinical characteristics and outcomes of ablation therapies (surgery and RAI treatment)  for GD over 15 years. The Authors performed a retrospective study and the study groups consisted of 10,991 GD patients at a single tertiary referral hospital between 2001 and 2015. The manuscript is clearly written and contains some important (however confirmatory) clinical information.

I have the following comments/suggestions:· 

  • The limitations of the study should be described in Discussion·        
  • Line 69-72 - One of the limitations is the analysis concerning Graves opthalmopathy (GO). The data concerning the severity and activity of GO are missing. It is possible that mild eye changes were missing. ·        
  • Line 76 – Is it possible to analyze the “immunological remission” – the concentration of TSH-R antibodies after ablation therapies·        
  • Line 131-137 – it would be also interesting to analyze subgroups of patients who underwent ablation therapy for GD, for example females in childbearing age, older patients with concomitant diseases. ·        
  • Table 2 – how many patients with GO received steroid prophylaxis during RAI treatment·        
  • Line 209-211 -   it is likely that “the increased rate of surgery for the treatment of GD reflects a robust increase in the incidence of thyroid cancer with the increasing number of high-resolution US in Korea. However, how do the Authors explain the decrease in RAI treatment? 

Author Response

Reviewer #2

Comment 1> Line 69-72 - One of the limitations is the analysis concerning Graves opthalmopathy (GO). The data concerning the severity and activity of GO are missing. It is possible that mild eye changes were missing. ·       

Response 1> Thank you for the critical comment. We re-analyzed the severity of GO through a review of patients’ medical records. GO was divided into mild and moderate-to-severe GO according to the European group on Graves’ Orbitopathy (EUGOGO) guideline. Among 89 patients with GO, 16 (5.1%) of them were classified into moderate-to-severe GO.

We added it in Table and Results section.

Revised manuscript, Materials and Methods 2.1, Line 69:

The presence of GO was identified based on the International Classification of Diseases, 10th revision code H06.2 and/or details of a visit to an ophthalmologist in our hospital for GO. The severity of GO was divided into mild and moderate-to-severe GO based on the European group on Graves’ Orbitopathy (EUGOGO) guideline [12].

Revised manuscript, Results 3.3., Line 143:

Although the prevalence of GO increased over time, the prevalence of moderate-to-severe GO did not differ (Table 1).

Revised manuscript, Results 3.4, Lines 161:

The prevalence of moderate-to-severe GO in the surgery for GD group was significantly higher than that in the RAI group (3.9% vs. 1.2%, p = 0.04). Among 12 patients with moderate-to-severe GO in the RAI group, 7 patients with active GO received steroid treatment to prevent the worsening GO.         

Table 2. Comparison of clinical characteristics among RAI and surgery groups.

RAI

(n = 968)

Surgery for GD

(n = 106)

Surgery for cancer

(n = 127)

Age at ablation therapy (years)

41 ± 13

39 ± 14

48 ± 14a, b

Sex, female, n (%)

663 (69)

91 (86)a

101 (80)a

Time from diagnosis

to ablation therapy (years)

4.9 ± 4.8

6.9 ± 6.9a

4.4 ± 6.0

Follow-up duration (years)

7.2 ± 4.4

5.5 ± 3.3a

7.2 ± 3.1b

GO, n (%)

67 (7)

10 (9)

12 (9)

   Moderate-to-severe GO, n (%)

12 (1.2)

4 (3.8)a

0 (0)

Reason for ablation therapy, n (%)

Intractable disease ± large goiter

820 (85)

100 (94)a

Adverse events of ATD

148 (15)

6 (6)a

Suspected thyroid cancer

127 (100)a, b

Data was presented as mean ± standard deviation or n (%).

RAI, radioactive iodine; GO, Graves’ ophthalmopathy; ATD, anti-thyroid drug

a p <0.05 compared to RAI group

b p <0.05 compared to Surgery for GD

Comment 2> Line 76 – Is it possible to analyze the “immunological remission” – the concentration of TSH-R antibodies after ablation therapies·       

Response 2> Thank you for the intriguing comment. After ablation therapy, 995 (83%) patients measured TSH receptor antibody at least once. Among 869 patients who achieved final remission, 390 (45%) patients have been proved to obtain the immunological remission, achieving negative conversion of TSH receptor antibody measurement. In detail, the RAI group showed higher rate of the immunological remission than that of the surgery group (40% vs 28% , p = 0.049). However, the follow-up duration was significantly longer in the RAI group than that in the surgery group (7.2 ± 4.4 years vs. 6.4 ± 3.3 years, p = 0.02). Therefore, comparing the immunological remission between RAI and surgery group is not fair in this retrospective study, and we did not include it on the present manuscript. The overall immunological remission rate is now described as below :

Revised manuscript – Materials and Methods 2.2., Line 78>

Remission of GD was defined as the achievement of euthyroid or hypothyroid status following the withdrawal of ATD for more than 6 months. Immunologic remission was assessed by the achievement of the negative conversion of TSH receptor antibody after ablation therapy. TSH receptor antibody was measured using a radio-receptor assay kit (RSR Limited, Cardiff, United Kingdom) [11].

Revised manuscript, Results 3.5., Line 186:

After ablation therapy, 995 (83%) patients measured TSH receptor antibody at least once. Among 869 patients who achieved final remission, 390 (45%) patients have been proved to obtain the immunological remission, achieving negative conversion of TSH receptor antibody measurement.

Comment 3> Line 131-137 – it would be also interesting to analyze subgroups of patients who underwent ablation therapy for GD, for example females in childbearing age, older patients with concomitant diseases. ·       

Response 3> Thank you for the interesting comment. Among GD patients who underwent ablation therapy, 41% (497/1,201) were women of childbearing age (20-44 years old). The proportion of women of childbearing age significantly decreased from 47% (2001-2005) to 36% (2011-2015) over time. The proportion of women of childbearing age in surgery for GD group (46%) was significantly lower than that in RAI group (59%) and surgery for cancer group (64%). There was no information about concomitant disease, so we cannot analyze a subgroup of older patients with concomitant diseases. We will refer to it for future study.

Revised manuscript, Results 3.3., Line 141:

The proportion of women of childbearing age (aged 20-44) significantly decreased to 47% (2001-2005), 40% (2006-2010), and 36% (2011-2015) (p for trend = 0.012).

Revised manuscript, Results 3.4., Line 158:

The proportion of women of childbearing age was significantly lower in the surgery for GD group than that in the RAI group (46% vs. 59%, p = 0.01).

Comment 4> Table 2 – how many patients with GO received steroid prophylaxis during RAI treatment·       

Response 4> Among GD patients who underwent RAI therapy, 67 patients had GO and 12 patients had moderate-to-severe GO. Among them, 7 patients with active GO received steroid prophylaxis whereas 6 patients with inactive stable GO did not. Usually, 15 mg of prednisolone was administered for 1-2 weeks. We added in the RESULTS section.

Revised manuscript, Results 3.4., Line 161:

The prevalence of moderate-to-severe GO in the surgery for GD group was significantly higher than that in the RAI group (3.9% vs. 1.2%, p = 0.04). Among 12 patients with moderate-to-severe GO in the RAI group, 7 patients with active GO received steroid treatment to prevent the worsening GO.

Comment 5> Line 209-211 - it is likely that “the increased rate of surgery for the treatment of GD reflects a robust increase in the incidence of thyroid cancer with the increasing number of high-resolution US in Korea. However, how do the Authors explain the decrease in RAI treatment?

Response 5> We appreciate the reviewer’s valuable comment. Patients considering ablation therapy chose RAI therapy or surgery. The proportion of patients who underwent ablation therapy decreased during the 15-year period, the proportion of patients who underwent surgery, especially surgery for suspected thyroid cancer, significantly increased. Therefore, the proportion of RAI therapy has decreased relatively. We revised the relevant content.

Revised manuscript, Discussion, Line 222:

During the 15-year period, the proportion of patients who underwent surgery within the ablation therapy group and the proportion of patients who underwent surgery for suspected thyroid cancer significantly increased. Compared to 2001-2005, the diagnosis of thyroid cancer increased in 2006-2015 with the technological evolution of high-resolution ultrasonography test in Korea [12,13]. Therefore, the proportion of RAI therapy has decreased relatively.